# Muscle Fiber Type Transitions with Exercise Training: Shifting Perspectives

**DOI:** 10.3390/sports9090127

**Published:** 2021-09-10

**Authors:** Daniel L. Plotkin, Michael D. Roberts, Cody T. Haun, Brad J. Schoenfeld

**Affiliations:** 1Health Sciences Department, CUNY Lehman College, Bronx, NY 10468, USA; danielplotkin96@gmail.com (D.L.P.); brad.schoenfeld@lehman.cuny.edu (B.J.S.); 2School of Kinesiology, Auburn University, Auburn, AL 36849, USA; mdr0024@auburn.edu; 3Fitomics, LLC., Pelham, AL 35124, USA

**Keywords:** fast-twitch fibers, slow-twitch fibers, strength training, endurance training

## Abstract

Human muscle fibers are generally classified by myosin heavy chain (MHC) isoforms characterized by slow to fast contractile speeds. Type I, or slow-twitch fibers, are seen in high abundance in elite endurance athletes, such as long-distance runners and cyclists. Alternatively, fast-twitch IIa and IIx fibers are abundant in elite power athletes, such as weightlifters and sprinters. While cross-sectional comparisons have shown marked differences between athletes, longitudinal data have not clearly converged on patterns in fiber type shifts over time, particularly between slow and fast fibers. However, not all fiber type identification techniques are created equal and, thus, may limit interpretation. Hybrid fibers, which express more than one MHC type (I/IIa, IIa/IIx, I/IIa/IIx), may make up a significant proportion of fibers. The measurement of the distribution of fibers would necessitate the ability to identify hybrid fibers, which is best done through single fiber analysis. Current evidence using the most appropriate techniques suggests a clear ability of fibers to shift between hybrid and pure fibers as well as between slow and fast fiber types. The context and extent to which this occurs, along with the limitations of current evidence, are discussed herein.

## 1. Introduction

Skeletal muscle contains a heterogeneous make-up of different fiber types that exist on a continuum of slow to fast, thereby making their function task-specific. Myosin heavy chain (MHC) isoform composition identification is a viable approach to classify fiber types along this spectrum. However, even amongst fibers of the same type, there are structural and functional characteristic differences; thus, MHC type is a somewhat crude but useful method to classify fiber types based on the relationship between MHC type and fiber function. In humans, type I, or slow-twitch, fibers possess slower twitch speeds and are relatively fatigue resistant. Type IIa fibers, or fast oxidative glycolytic (FOG) fibers, present higher twitch speeds than type I fibers but are less fatigue resistant. Type IIx fibers, or fast glycolytic fibers, possess the fastest twitch speeds but are highly fatigable [1]. These characteristics differentiate the respective utility of each fiber type in a variety of real-life contexts [2] and, thus, are an area of great interest in both healthy and diseased populations.

The ability of fiber types to transition from slow to fast and vice versa has been an ongoing topic of contention and investigation. It is well-established that an individual’s muscle fiber type composition can be relatively predictive of sports performance, with a greater proportion of type I fibers predictive of success in slower, longer distance events [3] and a greater proportion of type II fibers predictive of success in higher velocity, shorter duration events [4,5]. Thus, the extent and context of fiber type plasticity have major implications on sport and training. The purpose of this paper is to review techniques that have been commonly used to quantify fiber type distribution and highlight their respective strengths and weaknesses. Additionally, we non-systemically explore the literature to ascertain if, when, and how fiber type shifting occurs with different types of exercise training. Finally, we discuss the limitations of current research, implications for practical application, and future directions for study.

## 2. Fiber Type Identification Techniques

Different techniques exist to analyze skeletal muscle fibers, and each technique has strengths and limitations depending on the research question. As such, results should be interpreted in light of the technique and how it is applied. The current methods that exist fall into three broad categories: (i) muscle homogenate analysis, (ii) histological analysis of tissue sections, and (iii) the electrophoretic analysis of myosin heavy chain isoform expression from single fibers.

The analysis of crude muscle homogenates can be useful for extrapolating tissue characteristics, given that the homogenate contains a mixture of various muscle fibers [6]. In short, frozen tissue is homogenized using specialized buffers. Lysates are then prepared for sodium dodecyl sulfate-polyacrylamide gel electrophoresis (SDS-PAGE). Following SDS-PAGE, gel staining (e.g., Silver staining or Coomassie staining) is performed to identify tissue MHC isoform content by molecular weight. Readers are encouraged to refer to Roberts et al. for further details regarding this method [7]. In essence, if the analyzed tissue presents a profile that contains ~50% type I MHC protein, ~40% type IIa MHC protein, and ~10% type IIx MHC, the researcher may extrapolate that the muscle contains these corresponding percentages of type I, IIa, and IIx muscle fibers. However, as is discussed below, a significant proportion of muscle fibers can co-express MHC proteins, exemplified by the existence of I/IIa fibers, IIa/IIx fibers, and I/IIa/IIx fibers [8]. As a result, this technique lacks specificity in estimating the presence of hybrid fibers. This is an important limitation if the intent of a given research project is to examine how exercise training potentially affects fiber type shifting.

Analysis of muscle tissue sections is another commonly used technique for fiber typing. In short, the analysis can be achieved in a variety of manners. Historically, researchers would perform biochemical staining for oxidative enzyme content. Such stains have included, for example, NADH staining or SDH staining, where darker myofibers indicate a more oxidative phenotype [9]. While informative, such studies do not allow for the confident identification of type I versus type II fibers but rather provide information regarding the oxidative properties of a muscle specimen. As a contextual example, researchers have shown that short-term voluntary wheel running in rodents leads to darker NADH staining of the soleus and plantaris muscle when compared to non-exercise controls [10]. This exemplifies how the differentiation of fiber type based on biochemical metabolic staining alone was not possible. Historical investigations have also utilized biochemical methods that stain myofibers based on the pH lability of different myosin ATPase enzymes. This method allows for distinct staining patterns of type I, IIa, and IIx fibers. A drawback of these methods is that they require meticulous precision, given the relatively high number of solution changes during tissue processing. Moreover, staining quality (and data extrapolation from resulting stains) is highly influenced by subtle changes in the pH content of the different pre-incubation and incubation solutions [11].

A more recent, widely adopted, and highly reproducible method of fiber typing includes using antibodies to probe different MHC isoforms. In short, this technique involves relatively simple incubations, including: (i) media to block the section for non-specific binding, (ii) a solution containing primary antibodies to stain for the fiber types of interest (e.g., type I and IIa antibodies, where unstained fibers would be IIx fibers), and (iii) a solution containing secondary antibodies that are conjugated to different fluorophores (e.g., anti-type I-FITC and anti-type II-TRITC). Thereafter, a fluorescent microscope can be used to merge the RGB image suitable for muscle fiber typing. One drawback to this technique is that it lacks the sensitivity to confidently identify hybrid fibers [12]. Additionally, some studies have typically stained type I fibers while leaving type IIa and IIx fibers unstained. This form of staining generates a binary analysis of type I versus II fibers when, in reality, fiber types span a spectrum of pure and hybrid fibers, as previously discussed.

Single fiber typing is unequivocally the best method to delineate fiber type make-up from a biopsy specimen. Single fiber typing is also the only method that allows for mechanical and molecular analysis in a fiber type-specific fashion. Briefly, this method involves separating individual myofibers in a petri dish of digestion solution under a light microscope. After each fiber is isolated, they can be individually homogenized. Thereafter, individual homogenates can be analyzed through SDS-PAGE and Silver staining, as discussed above. Due to the small amount of protein within the sample, however, fiber pooling is sometimes needed. Additionally, separating and analyzing each individual fiber is tedious, making this technique very time and labor-intensive. An additional caveat is related to the potential of biopsies sampling different muscle fibers after an intervention versus before. The assumption is that by standardizing the anatomical location of a biopsy from pre- to post-intervention, a reasonable comparison can be made in the same region of muscle fibers. But, while standardizing the anatomical location of tissue sampling can help, the depth of the muscle from which the sample is extracted, small deviations from the original biopsy site, and other factors involved with tissue extraction and processing can influence the downstream analysis and inferences. In other words, it is possible that different single muscle fibers could be compared from pre- to post-study, and this could influence the inferences made about fiber type transition in response to an intervention. With that said, this limitation also exists for all the aforementioned techniques. It is also important to note that the number of fibers necessary to ascertain the extent of fiber type shifts has not been elucidated and may differ by individual, muscle, and muscle region. Nonetheless, single fiber isolation and back-end analysis allow for the confident identification of pure and hybrid fibers, and multiple studies discussed below have harnessed this technique to make unique discoveries in relation to how training can affect muscle fiber type. Figure 1 provides a summary of techniques for analyzing muscle fiber types.

## 3. Resistance/Sprint and Power Training

The current literature indicates that resistance training performed at slower speeds due to the use of relatively high loads (>70% of one-repetition maximum) produces a shift from IIx and IIx/IIa hybrids to more of a pure IIa phenotype and less shift in pure type I fibers, at least in the longitudinal timeframes that have been observed [13,14]. However, “power training” carried out at faster speeds generally shows somewhat less of a loss in IIx and IIx/IIa fibers and a concomitant decrease or shift in type I fibers to a faster phenotype. For example, Liu et al. compared changes in fiber type characteristics following combination training (i.e., the performance of ballistic and plyometric exercise in conjunction with strength training) versus maximal strength training in recreationally trained physical education students [15]. The authors used muscle homogenate gel electrophoresis to identify MHC isoforms of participants’ triceps brachii. In short, the authors reported that the group that completed 6 weeks of strength training alone experienced a shift to a more IIa phenotype (49.4% to 66.7%, *p* < 0.01) from IIx (33.4% to 19.5%, *p* < 0.01), with no change in type I fiber proportion. However, the combination training group did not display a significant loss of IIx fibers but instead experienced an increase in type IIa fibers (47.7 to 62.7%, *p* < 0.05) and a loss in type I fibers (18.2% to 9.2%, *p* < 0.05). Similarly, Malisoux et al. found that “stretch shortening cycle training” increased IIa phenotype (33.4% to 40.6%), as assessed by SDS-PAGE single fiber typing [16]. In contrast to Liu et al., Malisoux et al. observed a greater reduction in type IIx phenotype (7.0% to 2.6%), which may be explained by the use of single fiber biopsy analysis technique as compared to homogenate analysis used by Liu et al., a greater total volume of exercise completed, differing exercise history, or other factors. Overall, the few resistance training and power training longitudinal trials available using single fiber typing have shown a relative resistance to a shift from pure type I fibers compared to shifts from pure fast-twitch phenotypes.

Andersen et al. used SDS-PAGE single fiber analysis and found that 8 weeks of sprint training in male sprinters increased the proportion of type IIa fibers in their vastus lateralis (from 35% to 52%; *p* < 0.05), with a corresponding reduction in the percentage of type I fibers (from 52% to 41%; *p* < 0.05) [17]. This study suggests that sprint training may produce a bidirectional shift toward a IIa phenotype, with a larger shift from type I than is seen in traditional RT. MHC IIa/IIx fibers also experienced a pronounced shift (from 13% to 5%; *p* < 0.05). It is important to point out that the sprinters had just completed a 3-week period of full rest when the first biopsy was taken, and training ensued thereafter. Thus, although speculative, it is possible that the amount of IIa/IIx hybrid shifting would have been less pronounced if not for the period of detraining.

The totality of research suggests that sprint, power, and plyometric training can elicit a transition toward more of a IIa fiber type. However, individuals that begin this style of training with more IIx fibers may not experience as much of a transition. It is not currently possible to know what changes occurred over longer time frames and how much initial fiber composition is due to genetic influence with cross-sectional evidence; however, they can give hypothesis forming insights. One such extreme example that points toward type IIx retention was seen in an elite sprinter whose biopsy, as assessed by single fiber analysis, contained 24% pure IIx MHC. Given the longitudinal evidence, it is reasonable to suspect that fiber type shifting may occur to a lower extent the more well-trained an individual is using a certain training style. However, some data challenge this contention. In this regard, D’Antona et al. used single fiber isolation techniques to compare fiber type distribution of the vastus lateralis in five well-trained competitive bodybuilders relative to non-trained controls [18]. Bodybuilders possessed ~35% type I fibers, whereas controls possessed ~48% type I fibers (*p* < 0.05). Both groups possessed a statistically similar proportion of type IIa fibers (~45%); however, bodybuilders possessed a strikingly high proportion of type IIx fibers (~15%), whereas controls possessed just ~5% (*p* < 0.05). The authors noted that this finding was unanticipated, given that several shorter-term resistance training studies typically report a IIx → IIa fiber type transition. It remains plausible that intense resistance training programs can cause an eventual shift from a slow to fast phenotype over time, thus explaining the discrepancy between longitudinal and observational evidence. However, equally or perhaps more likely is that those with genetic predispositions retain/gain a higher proportion of the most advantageous MHC phenotype.

## 4. Endurance Training

Endurance training generally induces a fiber type shift toward a more oxidative phenotype. This is logical from the specificity of the training standpoint, considering the increased consumption of oxygen during endurance exercise. The rate of fiber type shifting appears to be less pronounced as endurance athletes progress deeper into event-specific preparation and if they begin an endurance training intervention with a higher abundance of type I fibers [19]. Luden et al. investigated the effects of 13 weeks of marathon training, followed by a 3-week taper on fiber type shifts in novice runners [20]. The investigators biopsied the vastus lateralis and soleus muscles and employed single fiber isolation techniques paired with SDS-PAGE for fiber typing. Vastus lateralis type I fiber composition increased (42.6% to 48.6%), I/IIa increased (5.1% to 8.2%), IIa decreased (40.1% to 35.8%), IIa/IIx decreased (11.9% to 6.4%), and IIx increased (0% to 1%). The soleus, a muscle comprised of predominantly type I fibers [21,22], experienced a similar but less pronounced shift, potentially indicating that certain muscles have a higher propensity for fiber type shifting than others. Importantly, biopsies were obtained after a taper, so it can be speculated that results would have been even more in favor of slower twitch abundance if measured directly post-training. Nevertheless, a shift toward type I fibers was apparent with higher volume endurance training. Other investigations have found similar patterns, indicating a shift toward type I fibers or a more oxidative phenotype after endurance training [23,24,25].

Multiple cross-sectional investigations have used one of the aforementioned techniques to demonstrate that well-trained endurance athletes possess a significantly higher proportion of type I fibers relative to sedentary or resistance-trained counterparts [26,27,28]. However, there is a dearth of long-term longitudinal evidence on fiber shifts in endurance athletes, precluding strong assertions on type I retention or shifts in elite athletics. However, one twin study has highlighted the potential for dramatic fiber type shifts over long periods of time. Bathgate et al. (Bathgate et al. 2018) carried out a case study in twins where one twin was largely sedentary throughout most of his adult life, while the other recreationally participated in endurance exercise for decades. Single fiber type analysis showed the trained twin had a vastus lateralis composition that was mostly slow-twitch (95% MHC I), amounting to 55% more MHC I fibers than the untrained twin. Moreover, the untrained twin possessed many more IIa and hybrid fibers than the trained twin. This study suggests that human muscle may dramatically shift toward one fiber type when training occurs over a lengthy period of time. Although further research is warranted to draw strong conclusions, it is possible that certain individuals have a greater plasticity/propensity toward fiber type shifts in one direction or the other and, perhaps, in general. While difficult to execute, studies examining multiple twin groups or appropriate cohorts could give insights into individual responses and the extent of fiber type shifts over a longer timeframe.

## 5. Disuse and Type IIx Fiber Overshoot

It is well characterized that fibers shift toward faster MHC expression during disuse [29]. However, there appears to be a type IIx overshoot after a period of training and detraining, which corresponds with superior fiber velocity characteristics [30]. Moreover, evidence in longer duration endurance events indicates that functional characteristics of the vastus lateralis are favorable after a taper in marathon runners, with no changes seen in functional characteristics of type I fibers and increased single fiber power and peak power in type IIa fibers post 3-week taper [25]. It is important to point out that even if functional characteristics of type I fibers remain unchanged, a taper may benefit an endurance athlete through fiber characteristics and shifts related to type II fibers. Such an effect may increase the capacity to generate enhanced force and power generation when running steeper hills or “kicking” at the end of a race. Indeed, it is well characterized that tapers are beneficial to a range of endurance athletes, and this may be an explanatory factor [31,32].

## 6. Potential Mechanisms

Early experimentation into mechanisms of fiber type shifts demonstrated the crucial role that innervation had on the profile of a muscle fiber [33]. In particular, research in cats found that cross-innervation of the soleus muscle with a nerve that innervated a muscle containing mostly fast-twitch fibers (flexor digitorum longus muscle, FDL) led to faster contractile twitch times. Additionally, the FDL muscle experienced slower twitch times following cross-innervation. However, subsequent experiments have yielded conflicting results with regard to how cross-innervation affects muscle fiber type. For instance, the cross-innervation of the sternohyoid and thyroarytenoid muscles in rats over 12 weeks has been shown to cause negligible fiber type shifting [34]. Thus, while input from alpha motor neurons likely affects fiber type, there are also other factors that may contribute to fiber type shifting as well.

Intra-myocellular signaling pathways induced by different forms of exercise likely play a large role in fiber type shifting. For instance, muscle activation increases calcium-activated signaling pathways, and these pathways contain signaling molecules, such as calcineurin, calcium/calmodulin-dependent protein kinases, and the nuclear factor of actived T cells (NFAT) transcription factor [35]. Interestingly, blocking the NFAT pathway has been shown to prevent a shift toward slow-twitch fibers [36,37]. Thus, the induction of NFAT via calcium, in particular, may be largely responsible for fiber type shifting during periods of exercise training.

AMP-activated protein kinase (AMPK) may be another signaling mediator involved with fiber type [38]. AMPK is a signaling mediator that exists in different isoforms and is typically associated with upregulation during endurance exercise, although there is evidence that resistance training also can increase its activation [39,40]. Interestingly, the inhibition of AMPK activity has been shown to blunt the fast-to-slow fiber type transition in rodents [41]. As previously mentioned, a hallmark adaptation that appears to occur rapidly during the first few months of endurance and resistance training involves the transition of IIx → IIa fibers. Thus, these signaling events may be critical to fiber type shifting in response to exercise training, given that repetitive muscle contractions during both endurance and resistance exercise bouts elicit increases in intracellular calcium along with the activation of AMPK.

Finally, it is also important to point out that genetic predisposition, in part, can explain some of the observed interindividual differences in fiber type make-up [42,43]. In this regard, the ACTN3 R577X genotype has been associated with muscle fiber type composition, where individuals harboring the RR genotype possess more type IIx fibers compared to those with the XX genotype [44]. Other researchers have shown that those with the angiotensin-converting enzyme gene (ACE) D allele had a 4.7% higher proportion of type I fibers than those with the ACE II genotype [45]. Interestingly, a recent study in endurance athletes (*n* = 103) and sprinters (*n* = 89) also identified five single nucleotide polymorphisms (PDE3A, PDE6C, and three non-annotated variants) that were associated with the predominance of type I muscle fibers [46]. A combination of genetic, training, nutrition, lifestyle, and perhaps other factors appear to interact and influence individual fiber type distribution [47,48]. Some of these modalities may induce shifts through overlapping mechanisms, while other mechanisms may be independent. Table 1 provides a summary of the methods of longitudinal exercise trials using single fiber analysis and their respective results.

## 7. Further Directions and Limitations

As alluded to above, a major limitation with studies that attempt to interrogate fiber type shifting with exercise training is a lack of longitudinal data in elite athletes. For example, elite sprinters and weight-lifters have a relatively high proportion of IIx fibers [4,5]; however, the preponderance of interventional evidence indicates a pronounced shift toward type IIa fibers. Thus, it remains questionable whether shorter-term findings (i.e., 6 to 16-week training studies) can be extrapolated to elite populations that undergo years or decades of training. It also appears evident that different muscles likely have a different propensity toward fiber type shifts, and there is a relative dearth of data in many muscle groups [12]. Evidence also suggests that different muscle regions (deep versus superficial and origin versus insertion) may have different fiber type patterns and, thus, likely display different shifting patterns in response to training [49]. In addition to fiber type differences that may be present between regions in response to training, it is also likely necessary to obtain multiple biopsies due to sample-to-sample fiber type variability. Horwath et al. obtained five biopsies along the VL of each participant’s leg (10 total biopsies) and found no pattern of difference in type II fiber distribution between biopsy sites along the muscle or between legs (average left 58 ± 8%, right 55 ± 8%). However, a notable within-subject non-systematic variability of 18 ± 4% was found. The researchers noted that they attempted to standardize sampling depth. Given such non-systematic variability, two or more biopsies per site would likely provide more robust findings when assessing fiber distribution [50]. However, from a practical perspective, we acknowledge that the biopsy technique is invasive, and such studies would be logistically challenging. Understanding fiber type shifts for different muscles and different regions using multi-site biopsy techniques holds promise in enhancing our ability to tailor training for different body parts/regions and for specific tasks. This area of research provides an exciting opportunity for discovery in the coming years.

## 8. Practical Applications

Based on our understanding of the current body of literature on muscle fiber types, we can draw the following conclusions from an applied standpoint:
Innate ability will affect how an individual performs at different sports/tasks, and fiber type composition likely plays an important role from a physiological perspective.Evidence suggests that muscle fibers have the ability to undergo fiber type transition, from hybrid to pure fibers, and between fiber types. The ability to discern hybrids is necessary to have a high degree of confidence in findings related to fiber type distribution.Given the dearth of evidence on the time course of long-term adaptation (>16 weeks), few long-term planning/periodization inferences can be drawn from the current fiber type transition literature. However, it likely would be prudent to maintain some of the force-velocity characteristics of the specific task/sport.Tapering practices are useful to allow for a transition back toward a faster fiber type. Some emerging evidence suggests an “overshoot” phenomenon in which type IIx fibers reach post-taper levels above what would be expected with rest alone. This may allow an athlete to be more explosive for a given event. Thus, tapering may present a viable strategy for power and endurance athletes to enhance performance.More research will help to elucidate differences in fiber type plasticity in other muscles, regional specificity, long-term adaptability, and other practical considerations, such as nutritional and training strategies.

## Figures and Tables

**Figure 1 sports-09-00127-f001:**
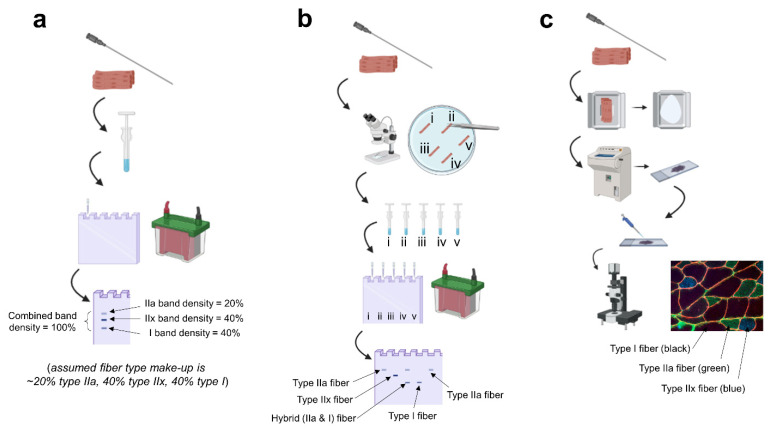
Summary of techniques. Legend: (**a**) fiber type estimation based on biopsied homogenates: a biopsy is obtained, the tissue is homogenized in specialized buffers and prepped for electrophoresis, and the gel is stained post-electrophoresis to visualize the percentage of each myosin isoform band. (**b**) singe fiber analysis: a biopsy is obtained, the tissue is teased apart under a stereoscope in a physiological digestion buffer, every single fiber is placed in a tube and homogenized, and electrophoresis is performed with back-end gel staining; this allows for the confident detection of hybrid fibers (example being “fiber iii”). (**c**) immunohistochemistry: a biopsy is obtained, the tissue is slow-frozen in a cryomold (or on cork) using freezing media, the frozen tissue is sectioned onto microscope slides using a cryostat, primary antibody solutions against various myosin isoforms are pipetted onto the slide, secondary antibody solutions against the primary antibodies are pipetted onto the slide, and the slide is mounted and imaged on a fluorescent microscope. Note, this image was generated using BioRender.com, and the fluorescent image is from the laboratory of MDR.

**Table 1 sports-09-00127-t001:** Longitudinal trials using single fiber analysis.

**Andersen, 1994**	Male Sprinters: N = 6	Sprint training preparation group, post-competitive season (no comparator)	The training was 2.5–3 h per day, 6 days a week, and consisted of a combination of strength and interval training for 3 months. Strength training was 2.5 days a week and consisted of exercises in the one to eight repetition range with 80–100% of 1 RM. Uphill sprints or interval-running were performed 2.5 days a week. One day was used for jumping exercises or recreational activities, such as basketball.	VL; Biopsies were taken post-3-week deload (which commenced after the track season) and post-intervention period.	Single Fiber Isolation. SDS-PAGE.	Mean changes: MHC I: 10.8% decrease (52.0+/−3.0 to 41.2+/−4.7) MHC I/IIa: 1.2% increase (0.2+/−0.2 to 1.4+/−1) MHC IIa: 17.6% increase (34.7+/−6.1 to 52.3+/−3.6) MHC IIa/IIx: 7.8% decrease (12.9+/−5.0 to 5.1+/−3.1) MHC IIx: 0.2% decrease (0.2+/−0.2 to 0.0)
**Andersen, 1994**	Male Soccer Players in the Danish National League: N = 14	High resistance knee extensor strength training group (ST): N = 8 Control group (no RT): N = 6	The training in the ST group was performed for 12 weeks, 3 times a week. Sessions consisted of four sets of eight repetitions of knee extensions using high resistance loads at velocities of 30–50 degrees per second.	VL; Biopsies were taken pre- and post-intervention.	Single Fiber Isolation. SDS-PAGE.	Mean changes: ST Group: MHC I: 3.1% increase (55.6+/−4.7 to 58.7+/−3.2) MHC I/IIa: 1.7% decrease (4.4+/−1.2 to 2.7+/−1.0) MHC IIa: 3.3% decrease (33.1+/−4.9 to 29.8+/−3.8) MHC IIa/IIx: 1.8% increase (6.7+/−3.2 to 8.5+/−2.2) MHC IIx: 0.1% increase (0.2+/−0.2 to 0.3+/−0.2) CTRL Group: All changes less than 3% except: MHC IIa: 6.2% decrease (27.7+/−5.2 to 21.5+/−6.8) MHC IIa/IIx: 3.9% increase (7.7+/−2.8 to 11.6 +/−4.9)
**Williamson, 2000**	Untrained Older Males: N = 7	Progressive resistance training (PRT) (no comparator)	The intervention was 12 weeks in duration, with training sessions 3 times per week. Training consisted of knee extensions at a cadence of 2–3 s for both the concentric and eccentric. The subjects performed three sets; the first two sets were 10 repetitions, and the last set was performed to volitional exhaustion. Nearly 2–3 min of rest was given between sets. All PRT exercise sessions were performed at 80% of one-repetition maximum (1 RM) and reassessed every 2 weeks in order to maintain prescribed intensity.	VL; Biopsies were taken pre- and post-intervention.	Single Fiber Isolation. SDS-PAGE.	The MHC I fiber distribution significantly increased by 10.4% after PRT, whereas the MHC IIa and IIx remained unchanged. Authors speculated the 12-week resistance training protocol might have not been sufficient in length to strengthen and increase the proportion of the denervated/reinnervated fibers often present in untrained older populations; thus, type II fibers would have been adapted at a slower rate than the type I fibers.
**Williamson, 2001**	Young Untrained: Male: N = 6 Female: N = 6	Progressive resistance training (PRT) group of men and women compared (same protocol, no comparator)	Same protocol as Williamson, 2000.	VL; Biopsies were taken pre- and post-intervention.	Single Fiber Isolation. SDS-PAGE.	Refer to Table 2 of Williamson, 2001 for a breakdown of the relative dominance of hybrid single fibers. Mean changes: Total hybrid fibers: 19% decrease in women, 29% decrease in men. MHC IIa increased 24% in men and 27% in women. Very little change in type I in either group.
**Widrick, 2002**	Untrained Males: N = 6	Resistance training (no comparator)	A total of 36 exercise sessions were performed at a frequency of 3 times per week on nonconsecutive days. Exercises for the lower body consisted of squats, knee extension, knee flexion, and calf raise. Upper body exercise consisted of bench press, lat pull down, shoulder press, triceps press, biceps curl, seated row, and abdominal exercise. Three sets of 5–10 of the exercises listed above (divided approximately equally between those targeting the upper and lower body). During all sessions, the training resistance was adjusted so that subjects were able to complete only the specified number of repetitions, plus or minus one repetition.	VL; Biopsies were taken pre- and post-intervention.	Single Fiber Isolation. SDS-PAGE.	Mean changes: Type I: 0% decrease (42–42) I/IIa: 4% decrease (4–0) IIa: 25% increase (30–55) IIa/IIx: 19% decrease (22–3) IIx: 3% decrease (3–0)
**Malisoux, 2005**	Untrained Males: N = 8	Stretch shortening cycle (SSC) exercise program	SSC exercise training program consisting of 24 sessions was performed 3 times per week for a total of 5228 jumps. Exercises included static jump, vertical countermovement jump, drop jump (height of 40 cm), double-leg triple jump, single-leg triple jump, double-leg hurdle jump, and single-leg hurdle jumps. The participants were instructed to perform all jumps at a maximal effort. The number of jumps was progressively increased during the first 4 weeks such that initial sessions were 20 min long and were 45 min at the end of the training period.	VL; Biopsies were taken pre- and post-intervention.	Single Fiber Isolation. SDS-PAGE.	Mean changes: MHC I: 0.8% decrease (30.0+/−4.9 to 29.2+/−4.1) MHC I/IIa: 3.1% increase (1.9+/−0.5 to 5.0+/−1.4) MHC IIa: 7.2% increase (33.4+/−5.2 to 40.6+/−4.2) MHC IIa/IIx: 4.7% decrease (26.9 +/−1.9 to 22.2+/−4.3) MHC IIx: 4.4% decrease (7.0+/−3.0 to 2.6 +/−1.9) MHC I/IIa/IIx: 0.4% decrease (0.8+/−0.4 to 0.4 +/−0.2)
**Trappe, 2006**	Relatively Untrained: Male: N = 4 Female: N = 3	Marathon Training (no comparator)	The training program was 4 days a week divided into two phases: a 13-week training period (10% increase in running volume per week), followed by a 3-week taper period. Compared with the last week of training (week 13), running volume was reduced by 25% in week 14, 47% in week 15, and 80% the week before the marathon.	Lateral gastrocnemius; Biopsies taken before the 16-week training plan, after 13 weeks of run training, and after 3 weeks of taper and marathon. Single muscle fiber MHC isoform experiments were only conducted before the training program and after the taper.	Single Fiber Isolation. SDS-PAGE.	Mean changes: Type I: 8% increase Type I/IIa: 5% decrease Type IIa: 2% increase Type IIa/x: 4% decrease Type I/IIax: 5% decrease Total hybrids: 11% decrease
**Luden, 2011**	Novice Runners: Male: N = 3 Female: N = 3	Marathon training (no Comparator)	Same protocol as Trappe, 2006.	VL, Biopsies were taken pre-training, following 13 weeks of run training, and again following 3 weeks of reduced training after a marathon (26.2 miles; 42.2 km). These time points are referred to as T1, T2, and T3, respectively.	Single Fiber Isolation. SDS-PAGE.	Soleus: T1 to T3 MHC I: 5.1% increase (71.1+/−6.1 to 76.2+/−4.7) MHC I/IIa: 0.3% decrease (6.4+/−1.7 to 6.1 +/−1.6%) MHC IIa: 4% decrease (21.4+/−4.8 to 17.4+/−3.9) MHC IIa/IIx: 0.8% decrease (1.1 +/−0.8 to 0.3) MHC IIx: Undetectable at both time points. VL: T1 to T3 MHC I: 6% increase (42.6+/−8.6 to 48.6+/−7.1) MHC I/IIa: 3.1% increase (5.1 +/−1.3 to 8.2 +−3.0) MHC IIa: 4.3% decrease (40.1+/−6.8 to 35.8 +/−4.4) MHC IIa/IIx: 5.5% decrease (11.9+/−2.9 to 6.4+/−2.4) MHC IIx: 1% increase (0 to 1+/−1.0

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
