# Peer review of "Muscle Fiber Type Transitions with Exercise Training: Shifting Perspectives"

_sports, 2021, doi:10.3390/sports9090127_

Round 1

Reviewer 1 Report

In this review, Plotkin and colleagues provide a brief overview of skeletal muscle fiber type composition, a discussion of various techniques for fiber type identification along with respective strengths and weaknesses, and then an evidence-based consensus on the possibility of fiber type switching. To conclude, the authors highlight limitations of current research and provide some recommendations for future directions in this area. Overall, I found the manuscript to be organized, well-written, and interesting, though not particularly novel.

  • Page 3 - An additional advantage of single muscle fiber SDS-PAGE is the ability to analyze various cellular and/or molecular parameters (e.g., contractile function) in a fiber type-specific fashion. Other fiber typing techniques do not present this possibility. This should be discussed. When discussing the single fiber SDS-PAGE technique it is also worthy to mention that the accuracy of this technique may depend on the number of muscle fibers assayed. Unfortunately, I do not believe the field has really reached a true consensus on this value, a task that is complicated by different muscles as well as fiber type characteristics of different populations.

  • Resistance/sprint and power training fiber type shifts (pages 3 & 4) – the authors provide a wealth of literature showing shifts from IIx of IIa/IIx to IIa, and rightfully highlight this as the most commonly observed phenomenon in this context. Meanwhile, the authors also discuss the possibility of a slow to fast fiber type shift (i.e., I à IIa), which I do not believe to be well supported. As evidence for this, the authors cite a study by Liu et al. which relied on homogenate from a muscle that is rarely studied in this context (i.e., triceps brachii). While there is cross-sectional data showing greater IIa and less I fibers in sprint and power athletes, this very well could be genetic. Thus, I encourage the authors to reconsider their stance on a slow to fast fiber type shift with resistance training. While certainly possible, I believe more longitudinal evidence is needed.

  • Based on patterns of use, some muscles contain predominantly one fiber type whereas others are more mixed. Might this influence fiber type shifts (i.e., do fiber type shifts occur in a muscle-specific manner)? The authors touch on this in the future directions and limitations section but I feel as though this needs to be brought up earlier for context in the resistance and aerobic training sections.

  • A section on aging, exercise, and muscle fiber type shifts is recommended as I believe it would add additional interest to the work.

  • Though not essential, I also believe a figure would greatly increase the impact of the work. Perhaps a graphic illustrating the different fiber typing techniques?

Author Response

Reviewer 1

In this review, Plotkin and colleagues provide a brief overview of skeletal muscle fiber type composition, a discussion of various techniques for fiber type identification along with respective strengths and weaknesses, and then an evidence-based consensus on the possibility of fiber type switching. To conclude, the authors highlight limitations of current research and provide some recommendations for future directions in this area. Overall, I found the manuscript to be organized, well-written, and interesting, though not particularly novel.

AUTHOR RESPONSE: Thank you for your thorough review. We have responded to each of your comments on a point-by-point basis. We hope our revisions meet with your satisfaction.

  • Page 3 - An additional advantage of single muscle fiber SDS-PAGE is the ability to analyze various cellular and/or molecular parameters (e.g., contractile function) in a fiber type-specific fashion. Other fiber typing techniques do not present this possibility. This should be discussed. When discussing the single fiber SDS-PAGE technique it is also worthy to mention that the accuracy of this technique may depend on the number of muscle fibers assayed. Unfortunately, I do not believe the field has really reached a true consensus on this value, a task that is complicated by different muscles as well as fiber type characteristics of different populations.

AUTHOR RESPONSE: Very good points, thanks. Changes have been made accordingly.

  • Resistance/sprint and power training fiber type shifts (pages 3 & 4) – the authors provide a wealth of literature showing shifts from IIx of IIa/IIx to IIa, and rightfully highlight this as the most commonly observed phenomenon in this context. Meanwhile, the authors also discuss the possibility of a slow to fast fiber type shift (i.e., I à IIa), which I do not believe to be well supported. As evidence for this, the authors cite a study by Liu et al. which relied on homogenate from a muscle that is rarely studied in this context (i.e., triceps brachii). While there is cross-sectional data showing greater IIa and less I fibers in sprint and power athletes, this very well could be genetic. Thus, I encourage the authors to reconsider their stance on a slow to fast fiber type shift with resistance training. While certainly possible, I believe more longitudinal evidence is needed.

AUTHOR RESPONSE: Thanks, good points about the relative dearth of evidence on type 1 fiber shifts, particularly in SCC, bodybuilding, and power type training. Additions were made to point out the relative resistance of type 1 fibers that seems to be observed in this context, at least in the periods analyzed. However, Andersen 1994 did show a substantial shift in type 1 fibers from sprint training, and the text was adjusted to underline that. Hope that clarifies our position.

  • Based on patterns of use, some muscles contain predominantly one fiber type whereas others are more mixed. Might this influence fiber type shifts (i.e., do fiber type shifts occur in a muscle-specific manner)? The authors touch on this in the future directions and limitations section but I feel as though this needs to be brought up earlier for context in the resistance and aerobic training sections.

 AUTHOR RESPONSE: Thanks this is an important point so we added a section alluding to this in the sections discussing cross sectional data both in endurance, and shorter duration/higher velocity events.

  • A section on aging, exercise, and muscle fiber type shifts is recommended as I believe it would add additional interest to the work.

AUTHOR RESPONSE: While a very interesting topic our hope was to present the longitudinal work as it pertained to training adaptations, specifically using single fiber methodology, as well as triangulating the training literature with cross sectional evidence. As there is only 1 study to our knowledge in older individuals using this methodology and other papers specifically focused on this topic, we are not sure it would add additional value.

  • Though not essential, I also believe a figure would greatly increase the impact of the work. Perhaps a graphic illustrating the different fiber typing techniques?

AUTHOR RESPONSE: Good suggestion. We have added a figure as requested.

Reviewer 2 Report

The paper is well prepared, and its theme is pertinent to the scientific community. However, throughout the paper, I suggest a better clarification regarding the classification of "pure" muscle fibres and hybrids. I recommend for this purpose the reading of the article "Pette, D., Staron, R. (2000). Myosin isoforms, muscle fibre types and transitions. Microscopy Research and Technique, 50: 500-509"

Also, in the article, I would like to see expressed the methodology used to prepare this review and the selection of the articles used.

Author Response

Reviewer 2

The paper is well prepared, and its theme is pertinent to the scientific community. However, throughout the paper, I suggest a better clarification regarding the classification of "pure" muscle fibres and hybrids. I recommend for this purpose the reading of the article "Pette, D., Staron, R. (2000). Myosin isoforms, muscle fibre types and transitions. Microscopy Research and Technique, 50: 500-509"

AUTHOR RESPONSE: Thanks, this is an important point and as such a section was added to the introduction to underline that fiber types exist on a continuum and even within fiber types there are differences that go beyond MHC and impact functionality.

Also, in the article, I would like to see expressed the methodology used to prepare this review and the selection of the articles used.

AUTHOR RESPONSE: Thanks. As this review was not conducted in a systematic fashion a short line was added to the introduction indicating this was the case.